# Comparison of the Estimation Ability of the Tensile Index of Paper Impregnated by UF-Modified Starch Adhesive Using ANFIS and MLR

Morteza Nazerian [1], Hossin Ranjbar Kashi [1], Hamidreza Rudi [2], Antonios N. Papadopoulos [3,*], Elham Vatankhah [1], Dafni Foti [3] and Hossin Kermaniyan [2]

1   Department of Bio Systems, Faculty of New Technologies and Aerospace Engineering, Shahid Beheshti University, Tehran 1983969411, Iran
2   Department of Bio Refinery, Faculty of New Technologies and Aerospace Engineering, Shahid Beheshti University, Tehran 1983969411, Iran
3   Laboratory of Wood Chemistry and Technology, Department of Forestry and Natural Environment, International Hellenic University, GR-661 00 Drama, Greece
*   Correspondence: antpap@for.ihu.gr

**Abstract:** The purpose of the present study is to offer an optimal model to predict the tensile index of the paper being consumed to make veneer impregnated with different weight ratios of modified starch (from 3.18 to 36.8%) to urea formaldehyde resin (WR) containing different formaldehyde to urea molar ratios (MR, from 1.16:1 to 2.84:1) enriched by different contents of silicon nano-oxide (NC, from 0 to 4%) using multiple linear regression (MLR) and adaptive neuro-fuzzy inference system (ANFIS) and compare the precision of these two models to estimate the response being examined (tensile index). Fourier-transform infrared spectroscopy (FTIR) and transmittance electron microscopy (TEM) were also used to analyze the results. The results of studying the adhesive structure using FTIR analysis showed that as the WR increased to the maximum level and MR increased to the average level (3%), more ether and methylene linkage forms due to cross-linking. TEM analysis also indicated that if an average level of silicon nano-oxide is applied, there will be more cross-linking due to the more uniform distribution and suitable interactions between the adhesive and nanoparticles. The modeling results showed that the ANFIS model estimates have been closer to the actual values compared to the MLR model. It can be concluded that the model offered by ANFIS has a higher potential to predict the tensile index of the paper impregnated with the combined adhesive of UF resin and modified starch. However, the MLR model could not offer a good estimate to predict the response. According to the preferred approach to predict the most effective property of resin coated paper, modelling would be useful to the research community and the results are beneficial in industrial applications without spending more cost and time.

**Keywords:** impregnated paper; tensile index; modified starch; nano silica; urea/formaldehyde resin; ANN; MLR

## 1. Introduction

The most common method to decrease formaldehyde release in wood products is to decrease the formaldehyde to urea molar ratio. However, it decreases the cross-linking and resin efficiency according to its mechanical strength consequently. Hence, in addition to decreasing the molar ratio, the combination of UF resin with other bio-based polymers such as starch can be used in the synthesis process. Using oxidized starch together with urea can make a hybrid system to produce wood composites without any formaldehyde release [1], while the bonding strength of the adhesive [2] and its strength improve [3,4].

The combination of nano-composite technology and decorative paper impregnating is a well-known way to modify and improve the properties of impregnated paper-based

veneers. Based on the studies conducted, the presence of Si–O–Al and Si–O–Si bonds was confirmed due to the presence of silicon nanoparticles in the starch polymer chain, while thermogravimetry analysis showed an improvement of the dimensional stability of starch adhesive, and a partial addition of the modified starch to resin improved the bonding strength in the plywood [5]. Nanoparticles are used widely to improve the mechanical, surface and visual properties of coatings, because many pores and gaps are filled [6]. Due to the amorphous nature of UF resin, the crystallinity of nanoparticles can decrease so that UF-nanofiller systems form [7]. According to Zheng et al. and Shi et al. [8,9], it became evident that adding nano-silica could improve the mechanical properties of resin. In addition, many of these nano-oxides have improved the resistance to abrasion and self-cleaning of some paints [9]. Therefore, it can be expected that nanomaterials can be also used as modifiers of cellulose and paper veneers [10]. However, due to their high surface energy, these materials tend to agglomerate [11]. It was showed that the usage of cellulose nanocrystals could improve mechanical properties such as tensile strength due to compatibility of nanoparticles with matrix [12,13].

To examine the effect of different independent factors on the responses being examined, different traditional and modern methods can be used. The regression analysis is used as a traditional tool to produce the model and is usually used to describe the quantitative relation between one response variable and one or more dependent variables by fitting the linear equation with the observed data [14]. The multiple linear regression (MLR) is the most common form of the linear regression analysis as a statistical method. MLR analysis is usually used to find the related coefficients in the model equations. The precision and correctness of the regression models improve using the MLR, while they decrease as the independent variables increase. Meanwhile, nonlinear dynamic modeling techniques such as ANNs and ANFS are powerful tools to solve complicated problems, control the quality, extract data and solve linear and nonlinear multivariate regression problems [15]. ANFIS is a scheme able to train ANNs to derive IF-THEN fuzzy rules using suitable fuzzy membership functions [16]. The advantage of ANFIS compared to ANN is the production of IF-THEN rules that can be described linguistically. The models offered by ANFIS include the relation between the input and output data through releasing the fuzzy linguistic rules. However, ANNs work in a form of the related trained weights, while they consume more time [17].

Many researchers have evaluated some basic engineering properties of wood composites using different mathematical techniques including linear/nonlinear regression, artificial neural network, genetic programming, etc., and have shown that these methods are valuable tools to solve different problems in composite material engineering applications and optimizations [18–25].

Two saturation and impregnated stages are performed to produce melamine-veneers. Due to the high price of melamine formaldehyde, only the UF resin is used in the first stage. Then, MF resin is used for final coating [26]. Paper sheets impregnated by thermoset resins are usually used to improve the physicomechanical properties, decrease formaldehyde gas emission and provide surface protection of wood-based composite products such as MDF and particleboard [27], while veneer can also emit formaldehyde gas.

So far, no study has been conducted on the possibility of using modern optimization methods to predict the properties of impregnated paper-based veneers. Hence, due to some scientific reports on the applicability of different optimization methods in other contexts of wood products, it is necessary to apply a reliable method to estimate the strengths of paper-based veneer strengths quickly and correctly. In this study, MLR and ANFIS are used to provide the test specimens using the second-order design. Hence, the purpose of the present study is to develop a suitable model to estimate the tensile index of paper impregnated with the UF resin made with different formaldehyde to urea molar ratios (MR) containing different weight ratios of the modified chemical starch to UF resin (WR) mixed with different nano-silica percentages.

## 2. Materials and Methods

### 2.1. Nano Silica

Silicon nano-oxide was prepared from Merck Co. (Darmstadt, Germany) with the specific surface area 210 m$^2$/g, purity 99.5% and average dimensions of particles 20 nm.

### 2.2. Preparation of the Starch Adhesive

Natural corn starch was used to make the natural adhesive. After loading 20 g starch, 40 mL distilled water and 2 g Sodium hypochlorite (NaOCl, equivalent to 1.25 cc) in a 100 cc beaker and putting in on a heater at the temperature 30 °C, the mixture was mixed for 30 min as the pH was stabilized at 9.5 (if necessary, NaOH 0.5 molar was added). After mixing the mixture well and neutralizing its medium using sulfuric acid 20% (H$_2$SO$_4$) and mixing it at the temperature 30 °C for 10 min on the mixing heater, the resulting mixture was put into the falcons and was centrifuged for 20 min at 2000 rpm to separate its water. Afterwards, when the mixture was removed from the centrifuge, the extra water was removed and the starch deposited in the falcons was washed by distilled water for several times. Using the Buchner funnel equipped with a filter paper and vacuum pump, the extra water was removed and it was dried at the room temperature.

After putting the beaker containing 10 g modified starch inside the bain-marie bath, 20 mL HCl 0.5 mol (hydrochloric acid 2%) was added to the mixture drop by drop with a pipette. After increasing the bath temperature gradually and stabilizing the mixture temperature in the beaker at 65 °C and mixing the mixture, the adhesive became thicker in 12 min before it was thixotropic. When the mixture became thick and after removing the beaker containing the adhesive solution from the bath (pH = 1.3), the reaction mixture was neutralized by the solution NaOH 0.5 mol (pH = 7–7.5). After neutralizing it, the adhesion of the adhesive was evident visually and sensually. To achieve a suitable concentration, the temperature of the neutralized adhesive reached 90–95 °C for 10–15 min using the bain-marie bath. Finally, the adhesive mass was put in the freezer for 24 h to be frozen to grind it, increase its durability and facilitate its suitable distribution when adding to the reaction medium at the time of UF resin synthesis and after getting dried in the room temperature. After putting the mixture in the freeze dryer and freeze-drying it for 24 h, the starch adhesive was ground well in a ball mill for 30 min at 260 rpm. The main physical and chemical properties of the used materials are illustrated in Table 1.

**Table 1.** Main properties of chemical used materials.

| Material | Properties |
|---|---|
| Nano silica | Specific surface area: 210 m$^2$/g, purity 99.5%, mean dimensions 20 nm |
| NaOCl | Molecular mass: 74.44 g/mol, density: 1.11 g/mL, purity: 15%, pH: 11 |
| NaOH | Molecular mass: 39.99 g/mol, density: 2.13 g/mL, purity: 99.5%, pH: 13 |
| H$_2$SO$_4$ | Molecular mass: 98.078 g/mol, density: 1.834 g/mL, purity: 98.5% |
| HCl | Molecular mass: 36.46 g/mol, density: 1.2 g/mL, purity: 98%, pka: $-6.3$ |
| Urea | Molecular mass: 60.06 g/mol, density: 1.33 g/cm$^3$, purity: 40%, pH: $\approx 7$ |
| Formaldehyde | Molecular mass: 30.03 g/mol, density: 0.81.53 g/cm$^3$, purity: 37.5%, pH: 2.5 |

### 2.3. Preparation of the UF-Starch Adhesive

Based on the design of experiment (DOE) used (Table 2) and to distribute Nano silica uniformly in UF resin, since it was not possible for nanoparticles to distribute in resin during the ultrasonic process due to the high temperature and the possible resin coagulation, a certain amount of nano-silica was weighed first and added to formalin (pH = 3). After 30 min at the frequency 70 Hz, a suitable distribution of particles was achieved in the formalin solution. According to the formaldehyde to urea molar ratios inserted in the DOE, formaldehyde (according to formalin concentration percent 37.5%) and urea (according to the purity 47%) were weighed to make five types of UF resin. In all five different molar ratios, all first-stage urea was added to the system in proportion to starch to urea weight ratio, while the second-stage urea was replaced by the modified starch and added to the

reaction medium at the end of the synthesis period. A three-necked flask equipped with a condenser, alcoholic thermometer and pH-meter was used. Immersing the flask in the bain-marie bath containing herbal oil and stabilizing its temperature at 30 °C, the whole formalin containing nano-silica and then the first part of urea were loaded in the flask while being mixed by a magnetic stirrer. The complex temperature and pH of the compound was 7 at 30 °C. Adding several drops of NaOH 0.5 mol with the concentration 20%, the compound's pH increased to 8–8.5. Increasing the reaction temperature to 90–95 °C, the mixture was mixed for 15 min. Increasing the temperature from 30 to 90–95 °C, pH was measured two to four times, so that it decreased to 6–6.5 while mixing and heating the mixture. At this temperature, adding some drops of sulfuric acid 1% (to decrease the time it takes for the pH to be equal to 4–4.5 more quickly) and in 15 min, a homogeneous single-phase solution was created. Next, the condenser was removed from the flask and two other blocked necks were opened in order that the extra water evaporated at the temperature 90–95 °C for 10–20 min. Afterwards, since the pH was acidic, adding some drops of NaOH, the mixture was neutralized (pH = 7–7.5). At this pH, the reaction complex was remixed for 20–30 min at 60–70 °C and it was concentrated. Then, after putting the mixture inside a beaker and putting in a cold bath and decreasing its temperature to 50–60 °C, the resin mixture temperature decreased to 20–25 °C. The compound's pH showed a weak basic medium (adding a little NaOH if necessary). Then, based on the DOE, the starch replaced by the second stage urea was added to the mixture and was put on the mixing heater for 30 min at the temperature 55–60 °C in order to ensure that the starch was distributed uniformly and dissolved completely. Afterwards, the UF-starch adhesive was dried at the environment temperature. Finally, 2–3 mL ammonia 20% was added to the adhesive solution to stabilize it.

**Table 2.** Experimental design and response (tensile index, T.I.).

| Run | Coded Values | | | Actual Values | | | T.I. |
|---|---|---|---|---|---|---|---|
| | $x_1$ | $x_2$ | $x_3$ | MR | WR. | NC | |
| 1 | 1.68 | 0 | 0 | 2.84 | 20 | 3 | 42 |
| 2 | 1 | −1 | −1 | 2.5 | 10 | 1.5 | 26.1 |
| 3 | −1.68 | 0 | 0 | 1.16 | 20 | 3 | 28.6 |
| 4 | 0 | 1.68 | 0 | 2 | 36.8 | 3 | 47.7 |
| 5 | −1 | −1 | −1 | 1.5 | 10 | 1.5 | 10 |
| 6 | 0 | −1.68 | 0 | 2 | 3.18 | 3 | 22 |
| 7 | 0 | 0 | −1.68 | 2 | 20 | 0.477 | 11.1 |
| 8 | −1 | −1 | 1 | 1.5 | 10 | 4.5 | 30.8 |
| 9 | −1 | −1 | −1 | 1.5 | 10 | 1.5 | 12.5 |
| 10 | 0 | 0 | −1.68 | 2 | 20 | 0.477 | 11.2 |
| 11 | 0 | 0 | 1.68 | 2 | 20 | 5.52 | 26 |
| 12 | −1 | 1 | 1 | 1.5 | 30 | 4.5 | 38.4 |
| 13 | −1 | −1 | −1 | 1.5 | 10 | 1.5 | 13.7 |
| 14 | 1 | 1 | 1 | 2.5 | 30 | 4.5 | 35.1 |
| 15 | −1.68 | 0 | 0 | 1.16 | 20 | 3 | 22.8 |
| 16 | −1 | 1 | −1 | 1.5 | 30 | 1.5 | 30 |
| 17 | 0 | 0 | −1.68 | 2 | 20 | 0.477 | 10.1 |
| 18 | −1 | 1 | −1 | 1.5 | 30 | 1.5 | 32.2 |
| 19 | −1 | −1 | 1 | 1.5 | 10 | 4.5 | 30 |
| 20 | −1 | 1 | −1 | 1.5 | 30 | 1.5 | 27.9 |
| 21 | 0 | 1.68 | 0 | 2 | 36.8 | 3 | 48.9 |
| 22 | 0 | −1.68 | 0 | 2 | 3.18 | 3 | 26.4 |
| 23 | 0 | 1.68 | 0 | 2 | 36.8 | 3 | 47.5 |
| 24 | 1 | −1 | −1 | 2.5 | 10 | 1.5 | 25.6 |
| 25 | 1 | −1 | −1 | 2.5 | 10 | 1.5 | 25.6 |
| 26 | 0 | −1.68 | 0 | 2 | 3.18 | 3 | 28.9 |

**Table 2.** *Cont.*

| Run | Coded Values | | | Actual Values | | | T.I. |
|---|---|---|---|---|---|---|---|
| | $x_1$ | $x_2$ | $x_3$ | MR | WR. | NC | |
| 27 | 1 | 1 | −1 | 2.5 | 30 | 1.5 | 40.5 |
| 28 | 0 | 0 | 0 | 2 | 20 | 3 | 31.6 |
| 29 | 0 | 0 | 0 | 2 | 20 | 3 | 31.2 |
| 30 | 0 | 0 | 0 | 2 | 20 | 3 | 30.5 |
| 31 | 0 | 0 | 0 | 2 | 20 | 3 | 31 |
| 32 | −1.68 | 0 | 0 | 1.16 | 20 | 3 | 21.6 |
| 33 | −1 | 1 | 1 | 1.5 | 30 | 4.5 | 40.8 |
| 34 | 0 | 0 | 0 | 2 | 20 | 3 | 31 |
| 35 | 1 | 1 | 1 | 2.5 | 30 | 4.5 | 40 |
| 36 | 1 | 1 | 1 | 2.5 | 30 | 4.5 | 40.2 |
| 37 | −1 | −1 | 1 | 1.5 | 10 | 4.5 | 16.2 |
| 38 | −1 | 1 | 1 | 1.5 | 30 | 4.5 | 40.6 |
| 39 | 0 | 0 | 0 | 2 | 20 | 3 | 30.2 |
| 40 | 1 | −1 | 1 | 2.5 | 10 | 4.5 | 32.6 |
| 41 | 0 | 0 | 1.68 | 2 | 20 | 5.52 | 27.3 |
| 42 | 1 | −1 | 1 | 2.5 | 10 | 4.5 | 30.6 |
| 43 | 1 | 1 | −1 | 2.5 | 30 | 1.5 | 42 |
| 44 | 1 | −1 | 1 | 2.5 | 10 | 4.5 | 33.2 |
| 45 | 0 | 0 | 1.68 | 2 | 20 | 5.52 | 25 |
| 46 | 1 | 1 | −1 | 2.5 | 30 | 1.5 | 41.7 |
| 47 | 1.68 | 0 | 0 | 2.84 | 20 | 3 | 35.7 |
| 48 | 1.68 | 0 | 0 | 2.84 | 20 | 3 | 34.3 |

*2.4. Characterization of Complex by FTIR and TEM*

To examine the changes in the surface functional groups, Fourier-Transform Infrared (FT-IR) spectroscopy analysis was performed using the pelletized specimens. About 100 mg of potassium bromide (KBr) was mixed with 2 mg of the adhesive samples changed into flour with a perfect coagulation. The prepared specimens were scanned using a Thermo Scientific Nicolet 6700 FT-IR spectrometer (Thermo Fisher Scientific, Waltham, MA, USA) in the wave number range from 600 to 4000 cm$^{-1}$.

The Transmission Electron Microscopy (TEM) allows one to understand the internal structure, spatial distribution and distribution of nanoparticles inside the polymer matrix qualitatively and indicates the defective structure directly and visually. The analysis was performed by the transmission electron microscope at 80 kV, JEOL JEM-1230. The TEM specimens with the thickness 50–70 nm were prepared by ultramicrotomy of the specimens capsulated in the epoxy matrix.

*2.5. Preparation of the Impregnated Paper Test Specimens*

First, the special papers consumed in the industry of melamine veneer production with the density 60 g/m$^2$ and the dimensions 20 cm × 30 cm were impregnated in an experimental coater by the adhesive with the concentration 50% based on Table 2. After impregnating them, the veneers were exposed to thermal treatment in an oven for 5 min at 140 °C. After 72 h air conditioning in the air conditioning room, the test specimens were prepared with the dimensions 25 mm × 180 mm with three replicas for any treatment by a cutter. The machine direction tensile strength of the specimens (Equation (1)) was obtained in the tensile strength tester (Gotech GT-7010-China, D2EP micro-computer tensile strength tester) equipped with the load cell 5 kN and the loading speed 25 mm/min to the time of failure based on the TAPPI T-494om-01 Standard [28]:

$$T = \frac{F}{b} \tag{1}$$

where, $F$ is the force applied and $b$ is the specimen's width.

After determining the tensile strength, the tensile index of the specimens was computed by the following equation:

$$TI = 1000\left(\frac{T}{R}\right), \quad \text{N·m/g} \tag{2}$$

where, $T$ is the force applied divided by the specimen's width ($F/b$, tensile strength) and $R$ is the specimen's grammage after being impregnated.

### 2.6. Design of Experiment (DOE)

The specimens were prepared as the factorial test design according to the independent variables being used including F to U molar ratio at five levels ($x_1$: 1.16:1, 1.5:1, 2:1, 2.5:1 and 2.84:1), the modified starch to urea weight ratio at five levels ($x_2$: 3.18:96.82, 10:90, 20:80, 30:70 and 36.8:63.2) and silicon nano-oxide percentage at five levels ($x_3$: 0.477, 1.5, 3, 4.5 and 5.52%, based on resin dry weight) and the dependent variable "tensile index". For this purpose, the Design of Expert Software Ver. 13 was utilized using the response surface methodology (RSM).

### 2.7. Modeling
#### 2.7.1. The Multiple Linear Regression (MLR) Model

The MLR model is a technique to model the linear relation between two or more variables. MLR is based on the least squares. The model is fit in a way that the sum of the squares of differences in the values observed and predicted is the least [29]. Equation (3) offers an MLR model:

$$Y = a_1 X_1 + a_2 X_2 + a_3 X_3 + \cdots + a_n X_n + C \tag{3}$$

where $Y$ is the dependent variable, $X_1$, $X_2$ and $X_3$ are independent variables and $a_1$, $a_2$, $a_3$ are the regression coefficients and $C$ is the intercept.

#### 2.7.2. The Adaptive Neuro-Fuzzy Inference System (ANFIS) Model

ANFIS is a neuro-fuzzy system connecting the fuzzy logic system to the neural network and making an intelligent hybrid system and benefiting from both the fuzzy logic and neural network in a way that its efficiency is proved in many of the very precise models [30]. Using a given input and output data set, the ANFIS toolbox function makes a fuzzy inference system (FIS) in which the membership function parameters are verified using the back-propagation algorithm alone or combined by a least squares type of the hybrid method. The neuro-adaptive learning method works similarly to the neural networks and provides a training method for fuzzy modeling.

### 3. Results and Discussion
#### 3.1. Statistical Analyses

Table 2 shows the results of the paper tensile index as the output of applying the production conditions of the impregnated paper. Table 3 can interpret the sequential model sum of squares as a summary of some models being used. The model guessed by the software being used, i.e., quadratic vs. 2FI, has the sum of squares equal to 1070, the df 3, the mean squares 355, F-value of 40.2 and $p$-value < 0.0001. The quadratic vs. 2FI model was chosen due to the $p$-value < 0.05% meaning that the model is significant. The lack of fit of test is defined as an inaccuracy test with the effect of the $p$-value and F-value results. Then according to sources, the software suggested a quadratic model with the sum of squares 67.5, df 3, mean squares 13.5, F-value 1.66 and $p$-value 0.172. A quadratic source was chosen because it had the highest $p$-value or >0.05 (5%). Based on the summary statistics which is a summary of some models based on the $R^2$ value, the significance is quadratic with the standard deviation 2.97, $R^2$ 0.929, adjusted $R^2$ 0.912, predicted $R^2$ 0.884 and Press 546. The

data suggested by the model have requirements based on the standard deviation and have a low press. However, the $R^2$, adjusted $R^2$ and predicted $R^2$ have high values.

**Table 3.** Sequential model fitting for tensile index.

| Sequential Model of Squares | | | | | | |
|---|---|---|---|---|---|---|
| Source | Sum of Squares | Df | Mean Square | F-Value | *p*-Value | |
| Mean vs. Total | $4.31 \times 10^4$ | 1 | 43,100 | | | |
| Linear vs. Mean | 3040 | 3 | 1010 | | | |
| 2FI vs. Linear | 255 | 3 | 84.9 | 27 | <0.0001 | |
| Quadratic vs. 2FI | 1070 | 3 | 355 | 2.48 | 0.0741 | |
| Cubic vs. Quadratic | 26.1 | 4 | 6.53 | 40.2 | <0.0001 | Sug. |
| Residual | 310 | 34 | 9.11 | 0.717 | 0.586 | Alia. |
| Total | 47,800 | 48 | 997 | | | |
| **Lack of Fit Tests** | | | | | | |
| Linear | $1.39 \times 10^3$ | 11 | 126 | 15.5 | <0.0001 | |
| 2FI | $1.13 \times 10^3$ | 8 | 142 | 17.4 | <0.0001 | |
| Quadratic | 67.5 | 5 | 13.5 | 1.66 | 0.172 | Sug. |
| Cubic | 41.4 | 1 | 41.4 | 5.09 | 0.0308 | |
| Pure error | 268 | 33 | 8.13 | | | Alia. |
| **Model Summary Statistics** | | | | | | |
| | Std. d. | $R^2$ | Adjusted $R^2$ | Predicted $R^2$ | Press | |
| Linear | 6.13 | 0.648 | 0.624 | 0.574 | 2000 | |
| 2FI | 5.85 | 0.702 | 0.658 | 0.621 | 1780 | |
| Quadratic | 2.97 | 0.929 | 0.912 | 0.884 | 546 | Sug. |
| Cubic | 3.02 | 0.934 | 0.909 | 0.861 | 651 | Alia. |

Table 4 shows the ANOVA results to determine the significance of the design used. The ANOVA results show that the input linear parameters ($x_1$, $x_2$ and $x_3$) significantly affect the tensile index ($p < 0.01$). It should be noted that the increase in the tensile index is due to the increase in MR, MR and an average level of NC. Based on Table 3, it is observed that the quadratic model used is significant and can be used in the optimization process by the artificial neural network models. Meanwhile, the interactions of $x_1x_2$, $x_1x_3$ and $x_2x_3$ together with the square effects of each variable $x_2$ and $x_3$ ($x_2^2$ and $x_3^3$) have also significantly affected the response according to the *p*-values. Among different effects described in the Table and based on the F-value, it is observed that the direct effect of $x_2$ has the highest effect (with the value 210) on the level of the response being examined. The non-significance of the lack of fit value also shows the suitable fit among the data.

**Table 4.** Response model and statistical parameters obtained from ANOVA for central composite design.

| Source | Sum of Squares | Df | Mean Squares | F-Value | *p*-Value |
|---|---|---|---|---|---|
| Model | $4.36 \times 10^3$ | 8 | 545 | 62.2 | <0.0001 |
| $x_1$-MR | 591 | 1 | 591 | 67.5 | <0.0001 |
| $x_1$-WR | $1.84 \times 10^3$ | 1 | $1.84 \times 10^3$ | 210 | <0.0001 |
| $x_3$-NC | 609 | 1 | 609 | 69.5 | <0.0001 |
| $x_1x_2$ | 39.9 | 1 | 39.9 | 4.56 | 0.0391 |
| $x_1x_3$ | 151 | 1 | 151 | 17.3 | 0.000171 |
| $x_2x_3$ | 63.5 | 1 | 63.5 | 7.26 | 0.0104 |
| $x_2^2$ | 195 | 1 | 195 | 22.3 | <0.0001 |
| $x_3^2$ | 573 | 1 | 573 | 65.4 | <0.0001 |
| Residual | 341 | 39 | 8.76 | | |
| Lack of Fit | 73.1 | 6 | 12.2 | 1.5 | 0.209 |
| Pure Error | 268 | 33 | 8.13 | | |
| Cor Total | $4.7 \times 10^3$ | 47 | | | |

### 3.2. Characterization Analysis

The FTIR spectrum of the (500–4000 cm$^{-1}$) natural starch (NS), modified starch (MS), combination of UF resin and modified starch MR1.5WR10 (F to U molar ratio and starch to urea weight ratio equal to 1.5 and 10%, respectively) and combination of UF resin and modified starch MR2.5WR30 (F to U molar ratio and starch to urea weight ratio equal to 2.5 and 30%, respectively) are given in Figure 1. The band 3000 cm$^{-1}$ to 3800 cm$^{-1}$ has intensified as WR and MR increased, showing the presence of more free hydroxyl groups due to the scission of chain in the alkaline oxidation process. Hence, this occurrence can be due to the reaction of NaOH with OH-groups forming the –O$^{-}$Na$^{+}$ group and rupture of hydrogen bond structure supplying the stability of the starch molecule and exposing the OH-groups [31]. When starch is treated, the intensity of the OH– absorption peak near 3285 cm$^{-1}$ decreases due to the decrease in OH-content after oxidation. However, the peak intensity of the UF-MS mixture has increased due to the increase in the MR. Furthermore, in this peak range showing the overlapping of the modified starch OH-groups and N-H stretching of primary amide in the UF resin, it is observed that as U is replaced by more modified starch, the peak intensifies, showing that even as the modified starch weight ratio increases, higher U to F molar ratios can be used. It means that even using higher levels of the modified starch, the peak intensity decreases as the U to F molar ratio increases due to the formation of more mono-, di-, and tri-hydroxymethyl urea [32] that interacts more with aldehyde groups chemically. Hence, it can be expected that as the starch-weight ratio increases in resin containing more urea, better properties can be expected from the adhesive.

The double peak in the range from 2925 cm$^{-1}$ to 2880 cm$^{-1}$ belonging to the symmetric and asymmetric stretching vibration of –CH$_2$ and –CH has increased the –CH and –CH$_2$ polarity, increasing the WR and MR in the presence of -O$^{-}$Na$^{+}$ with respect to the increment in absorption peak. When decreasing the WR and MR, the –CH and –CH$_2$ absorption peaks decrease. The weak C–H bond is related to the increase in the U to F molar ratio as the WR decreases. Changes in this peak related to starch, modified starch and adhesive containing a low F to U molar ratio and weight ratio follow a similar trend. However, as the WR ratio and MR increase, the peak intensifies that can be related to the increase in the ether and methylene linkage in the coagulated adhesive and the resulting cross-linking. Therefore, it can be said that using more starch and increasing the U to F molar ratio, polycondensation can occur due to copolymerization and partial intermolecular hydrogen bonds.

In the band 1645 cm$^{-1}$ belonging to the stretching vibration band of C=O group of amide I and II and amide I N-H scissors [33], it is observed that the peak intensity has increased after the oxidation, showing that more aldehydes are formed in the oxidation process. Simultaneously, as more modified starch is consumed, the peak intensity reaches the maximum level as the MR is maximized. According to the curve (b), it is observed that in practice not only MR but also the increase in the WR have increased the aldehyde group in the adhesive. Hence, the disappearance of stretching of the specimens with more U to F molar ratio shows the possibility of the reaction between aldehyde and carboxyl groups on the amylose and amylopectin chain of the oxidized starch and –NH and –NH$_2$ groups to form the UF-MS adhesive through the polycondensation reaction [2]. The overlapped peaks at 1527 cm$^{-1}$ formed in the specimens containing the modified starch are related to the N-H bending vibrations of amide II intensified as the MR has increased. However, no new peak is observed in the modified starch after the oxidation and only the peak has intensified as F to U ratio is increased. The peak at 1385 cm$^{-1}$ showing the C–H stretching and bending modes of methylene in the UF specimens containing the modified starch resulting from the shift in the peak at 1337 cm$^{-1}$ in the starch and the modified starch [34] has also intensified, indicating ester COO–R- group in both UF-MS adhesives. The absorption band at 1240 cm$^{-1}$ shows the presence of C–O–C bonds in UF-MS [35] while as the molar ratio has increased, the peak width has increased, showing the formation of asymmetric stretching vibration in the form of ester connection with –COOH in the formed glucose as the U to F molar ratio has increased due to the presence of –OH on dimethylol urea. However, it was observed that this peak in the pure UF resin does not exist normally [33]. The band 1145 cm$^{-1}$ related to the

C–N stretching shows a significantly lower intensity in the modified starch. In addition, its intensity in the UF-MS adhesive with a lower formaldehyde molar ratio has decreased less. However, as the modified starch is added and the F to U molar ratio has increased, the peak has disappeared, showing that it can be happen the polycondensation reaction of –COOH of the modified starch with urea nitrogen to form U-OS.

Ether groups can be identified by the presence of the peaks related to the C–O–C bonds at 1050 cm$^{-1}$ in the glycosidic ring [36]. The presence of the peak of the C–O–C bond in more intense spectra in starch and modified starch and in the decreased intense in modified starch + UF with less formaldehyde specimen and its absence in starch + UF resin with more formaldehyde specimen show the chemical interaction to graft starch in the copolymerization process with UF. NaOH can make the starch expose more hydroxyl reduction ends by swelling starch granules and forming O–Na$^{+}$- groups. Under the synthesis conditions such as high shear stress and temperature, NaOH reacts with starch molecule hydroxyl groups and gives a molecule containing di-carbonyl group through rearrangement of the molecule [31]. In addition to the oxidation of the starch hydroxyl groups, NaOH weakened the absorption peak of C–O–C bond of the starch molecule in the range 990 cm$^{-1}$ in oxidized starch and also in the specimens containing the UF resin with a lower formaldehyde molar ratio. However, it has increased the peak intensity in the range 600–800 cm$^{-1}$ for the specimen with more modified starch and resin with more formaldehyde. It shows that NaOH could create the unsaturated connection resulting from the broken glycoside links containing heterocyclic rings and carboxylic acid molecules so that due to the cross-linking reaction, the band 860 cm$^{-1}$ showing di-carbonyl groups disappeared in the adhesive specimens containing UF.

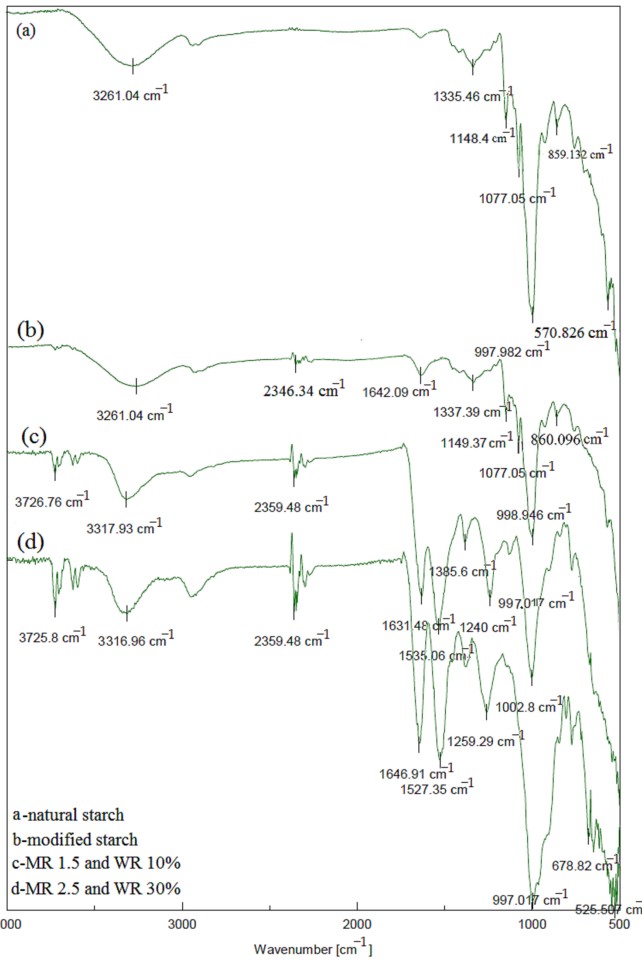

**Figure 1.** (**a**) FT-IR spectra of natural starch, (**b**) modified starch, (**c**) UF-MS (MR of 1.5 and WR of 10%) and (**d**) UF-Ma (MR of 2.5 and WR of 30%).

Figure 2 shows TEM photos of the glue line containing nano-silica. Adding 1.5% nano-silica has led to the suitable distribution of the nanoparticles in the matrix. The distance between platelets of nanoparticles has increased and the polymer chain entered the platelets space. This nanostructure is related to the flower-like unified structure [7] giving better properties to the nanocomposite that is due to the individual separation of nanoparticle layers in the continuous polymer matrix by the average distance which depends on the nanoparticle loading (exfoliation). The comparison of the Figure 2a,b indicates that as the nanoparticles loading increases to 4%, the intercalation degree of the nanoparticles' layers in the UF resin polymer matrix that is not an ideal structure for a nanocomposite is accompanied by huge aggregation or tactoids and does not offer nanocomposites with better properties like an exfoliated structure. Based on the results obtained, it became clear that increasing nanoparticles to 3% resulted in a significant increase in the strength. However, as more nanoparticles were loaded in the polymer matrix, the strength did not increase. This phenomenon can be related to the difficulty in the uniform distribution of nanoparticles in the matrix at higher levels of nanoparticles loading.

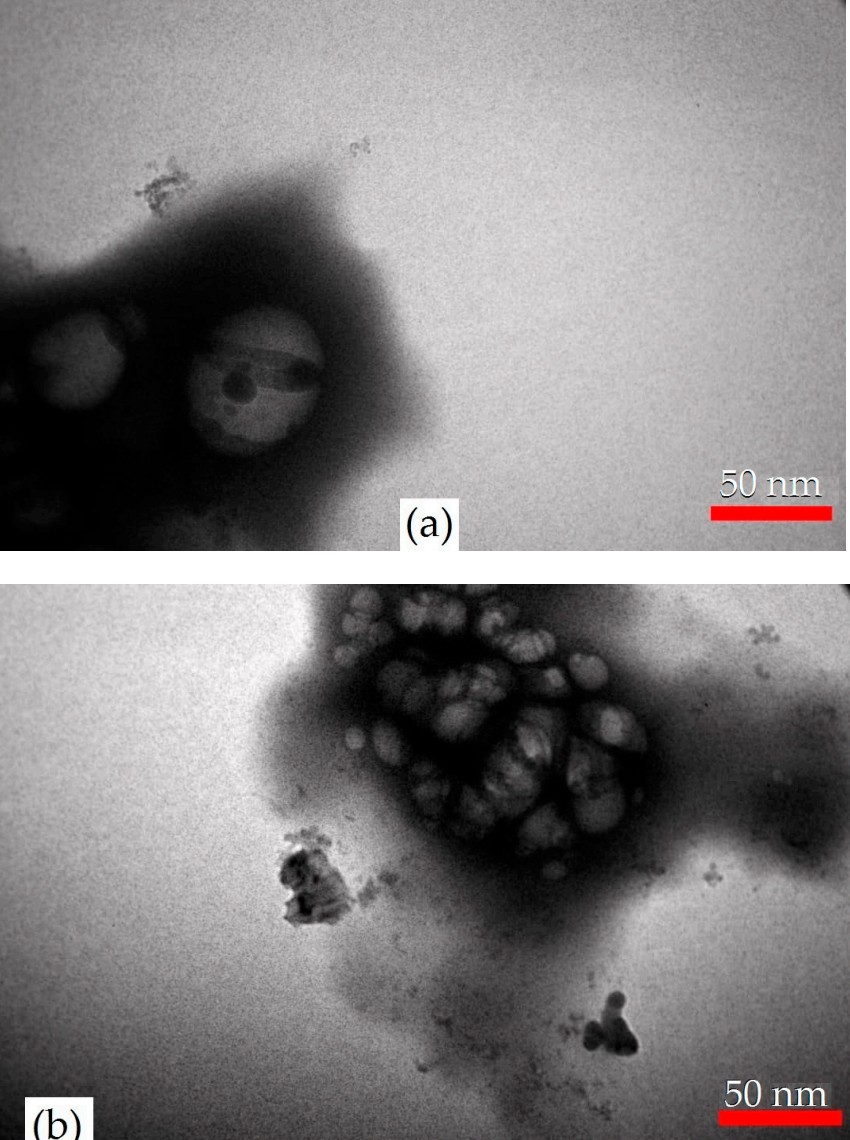

**Figure 2.** Transmission electron microscopes photographs of UF resin mixed with 1.5% (**a**) and 4% (**b**) nano SiO$_2$ particles.

### 3.3. Evaluation of the Models

The diagram of the general block of ANFIS is composed of three inputs and one output as shown in Figure 3. The output parameter or the paper's tensile index evaluating the mechanical efficiency depends on three input parameters, i.e., MR, WR and NC.

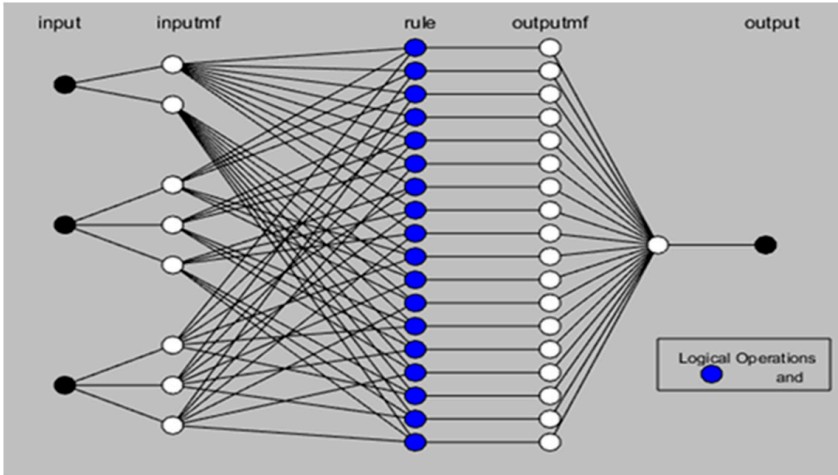

**Figure 3.** Optimized structure of the ANFIS model.

ANFIS models were developed using the fuzzy logic toolbox of MATLAB R2015b software. Sugeno fuzzy reasoning was used with two Gaussian membership functions for each input. A defuzzifier linear formula was used in the output of each rule; and the total output is the weighting average of each rule output. The structure of rules and the tuned FIS are shown in Figure 3, including 18 rules with the logical connector AND for all rules.

To develop the ANFIS models to predict the strength of the impregnated paper, the available data sets obtained by the experiments were used including 48 input vectors and the corresponding output vectors. These data sets were determined randomly as the training set. After training, the fuzzy inference calculations of the developed model were performed. Then, to evaluate the performance of the ANFIS predictions, five input vectors resulting from the test data sets were presented to the trained network and the predicted network responses were compared with the experimental responses.

The effect of the developed MLR and ANFIS models to predict the response was evaluated by determination coefficient ($R^2$), root mean square error (RMSE), mean absolute error (MAE) and Sum of Square Error (SSE). The results are presented in Table 5. $R^2$ must be close to 1 for a good relation between the predicted and actual values. It was observed that the model offered by ANFIS has much higher and more satisfactory $R^2$ values together with a good fit of the model [37] (0.9612 for the training data sets and 0.9261 for all data sets) compared with the values obtained by the MLR model both for the training data sets (0.5174) and all data sets (0.4818). MSE is close to the line fit with the data points and based on it, RMSE was calculated for the model including the square root of the MSE. The RMSE of the ANFIS was much lower than the model offered by MLR, showing and supporting the good fit of the model. MAE and SSE show the precision and correctness of the model [38]. The lower values of these statistical indices show the better performance of the model. They were determined for the developed models with their values given in Table 5. Based on the statistics, the ANFIS model was much better than the MLR model. It can be observed that the ANFIS model has been more precise according to the higher error values of MAE and SSE for the training data sets (1.2942% and 112.22, respectively) and all data sets (1.5812 and 355.96, respectively) compared to the MLR for the training data sets (14.04% and 14,557, respectively) and all data sets (14.95% and 16,727, respectively).

**Table 5.** Comparison of statistically criteria of models.

|  | Training Data Set | | All Data Set | |
|---|---|---|---|---|
|  | **MLR** | **ANFIS** | **MLR** | **ANFIS** |
| $R^2$ | 0.5174 | 0.9612 | 0.4818 | 0.9261 |
| RMSE | 17.48 | 1.8168 | 18.66 | 2.7232 |
| MAE | 14.04 | 1.2942 | 14.95 | 1.5812 |
| SSE | 14,557 | 112.22 | 16,727 | 355.96 |

The plots of the experimental and predicted values are drawn in Figure 4 supporting the observed high $R^2$ values (Table 5) for both models. The $R^2$ value must be close to 1 in order that the good relation between the experimental and predicted values is proved. Additionally, both models showed a considerably different $R^2$ value so that the value obtained by the ANFIS model is closer to 1 compared to the MLR model, indicating the more suitable estimate of the ANFIS model to predict the response compared to the MLR model. As it is observed, almost all data are scattered around the regression line, showing that only 8% of all variations are not described by the developed ANFIS models.

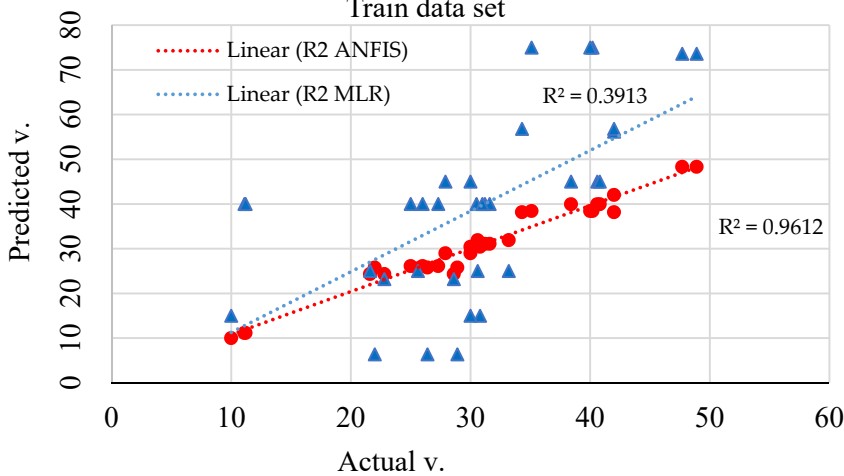

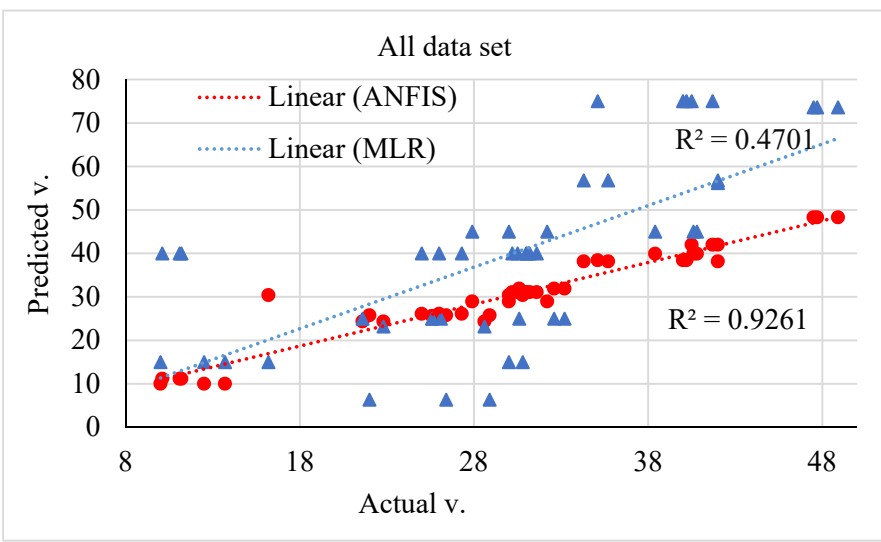

**Figure 4.** Correlation between the actual data of tensile index (MPa) and predicted ones by ANFIS and MLR models in both training and all data (blue triangles and red circles marks belong to MLR and ANFIS models, respectively).

In Figure 5, the residual distribution of both optimization techniques is depicted. The variation in residuals is completely different for both methods so that the error values are large for the MLR and they are very low for ANFIS and show a low variation compared to the MLR. It shows that the MLR presents a regression equation that could not predict the response being examined appropriately and states the effect of the experimental parameters and their interaction on the response with a lower precision compared to the ANFIS. ANFIS gives an opportunity to adapt to the experimental design to make the model. The ANFIS approach has been flexible and can add new data to make a reliable model; therefore, it is more reasonable and stable to interpret the tensile index of the papers through the ANFIS architecture process.

Superiority of ANFIS approaches toward estimation of response can be proved according to other statistical indicators so that the experimental values are offered like the estimated values of both models against the number of runs. The distribution of differences between the residual error values estimated by the ANFIS and MLR can be observed in Figure 6. It is observed that the variation in the values predicted by the ANFIS model is much less than those predicted by the MLR model. It means that the experimental data are fit with a high precision using the ANFIS model compared to the MLR model and the prediction variation of the ANFIS approach is less than the other. ANFIS integrates both advantages of the fuzzy logic and neural network techniques to model data. Artificial intelligent methods such as ANFIS, etc., are able to train to estimate the nonlinear functions while other methods can only be limited to special models such as second-order polynomial.

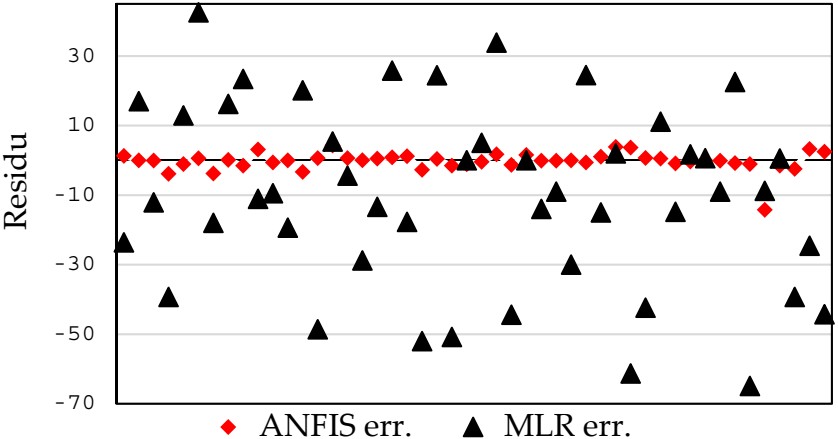

**Figure 5.** Comparison between the residual error distribution by ANFIS and MLR.

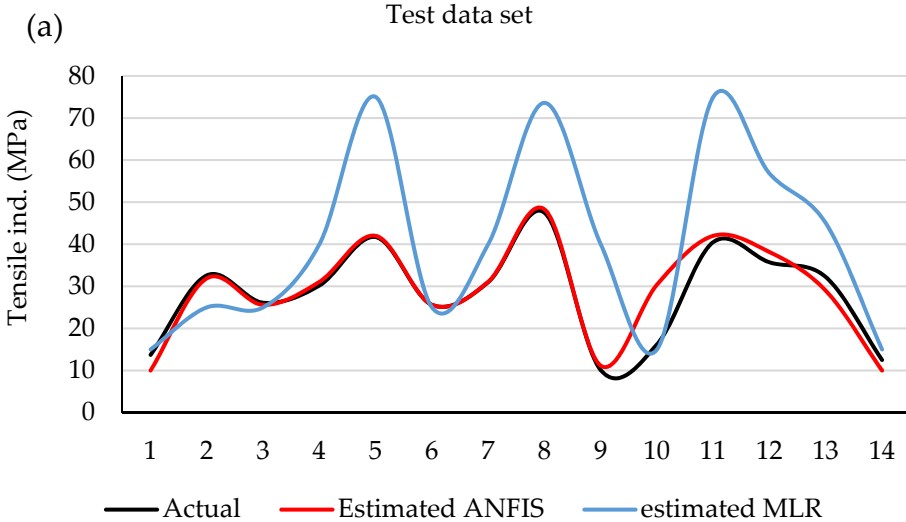

**Figure 6.** *Cont.*

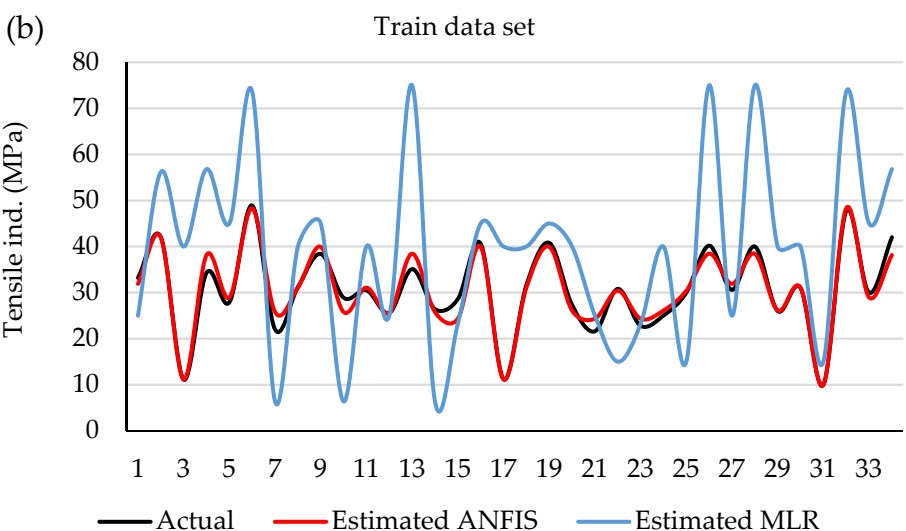

**Figure 6.** Comparison of experimental and predicted values of test (**a**) and train (**b**) data sets by the ANFIS and MLR models for each experimental run.

The fuzzy inference diagrams of the ANFIS model with 18 rules are presented in Figure 7. Accordingly, the response being examined can be affected by changing the input variables. Using the middle level of MR, maximum WR of adhesive containing the middle level of nano-silica percentage, the response becomes the maximum, reaching 48.3 MPa. In other words, matching the ANFIS output with the experimental tensile index, the prediction accuracy of response was proved based on the ANFIS model. The 3D diagrams show the interaction of the independent variables being examined (Figure 8) and the level of the response dependency to the changes in the variables so that the changes in the response is finally similar to the final output calculated in Figure 8. In the interaction of MR and WR, the maximum strength is where the values of both factors are at the maximum (Figure 8a). However, the strength will be at the maximum if the MR is at the maximum in the interaction of MR and NC, while the nano percentage is at the middle level (2%) (Figure 8b). The interaction of WR and NC also shows that as the WR increases and the nano-silica becomes at the minimum, the tensile index will be at the maximum (Figure 8c).

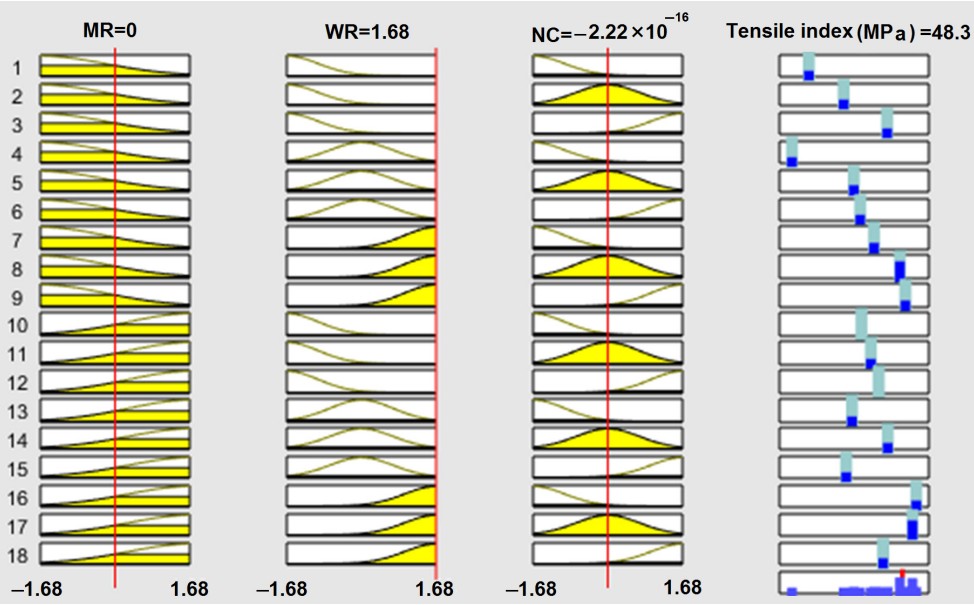

**Figure 7.** Rule viewer of FIS model for training data (yellow colors and curves show the optimum value of every input and used function, respectively, in each rule).

When applying the quadratic model to design the experiments and describe the interaction of the factors being examined, the model can be used to describe the direct effect of all three factors. Hence, curves are formed showing the actual effect of the factors on the response.

At the same time, using the ANFIS and MLR models, the precision of the estimate of the effect of each independent variable being examined can be measured by comparing them with the actual affecting values. Figure 9 compares the tensile index estimated by the ANFIS and MLR models with its actual values. It is observed that the effects of the independent variables MR, WR and NC on the response estimated by the ANFIS and MLR are distinct from each other so that the responses estimated by ANFIS are very consistent with the actual responses. However, the responses estimated by the MLR model are not consistent with the actual data. It is also evident based on the statistics such as $R^2$, RMSE, MAE and SSE (Table 5). It is observed that the effect of WR is more than other factors according to the slope of the tangent on the related curve, showing that WR could affect the response more than other factors. Based on the ANOVA analysis (Table 3), F-values of WR (210) are much more than those of other terms whether in the direct, interactive or square effects of the factors, showing the much higher effect of this factor though other factors also had significant effects. It is observed that as WR increases continuously, the tensile index increases continuously. However, it is observed that although the increase in MR has increased the tensile index continuously, the intensity of the increase is very slow. It is also evident in the F-value in the ANOVA table. It is observed that the direct effect of nano-silica has been almost at the middle level of the maximum consumption and its effect on the tensile index is negative whether before or after it.

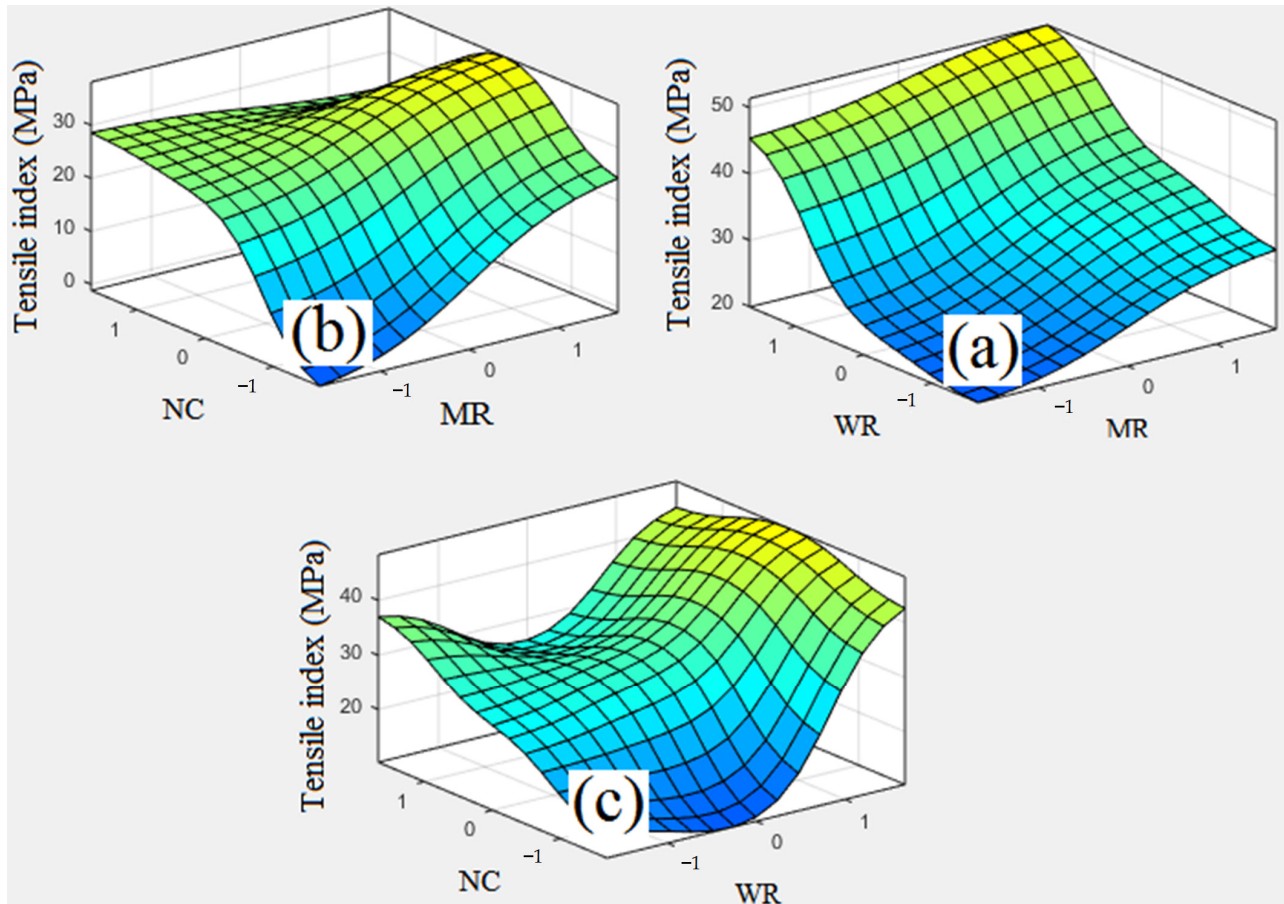

**Figure 8.** The interaction of the molar ratio × weight ratio (**a**), molar ratio × nano percentage (**b**), and weight ratio × nano percentage (**c**) on the tensile index.

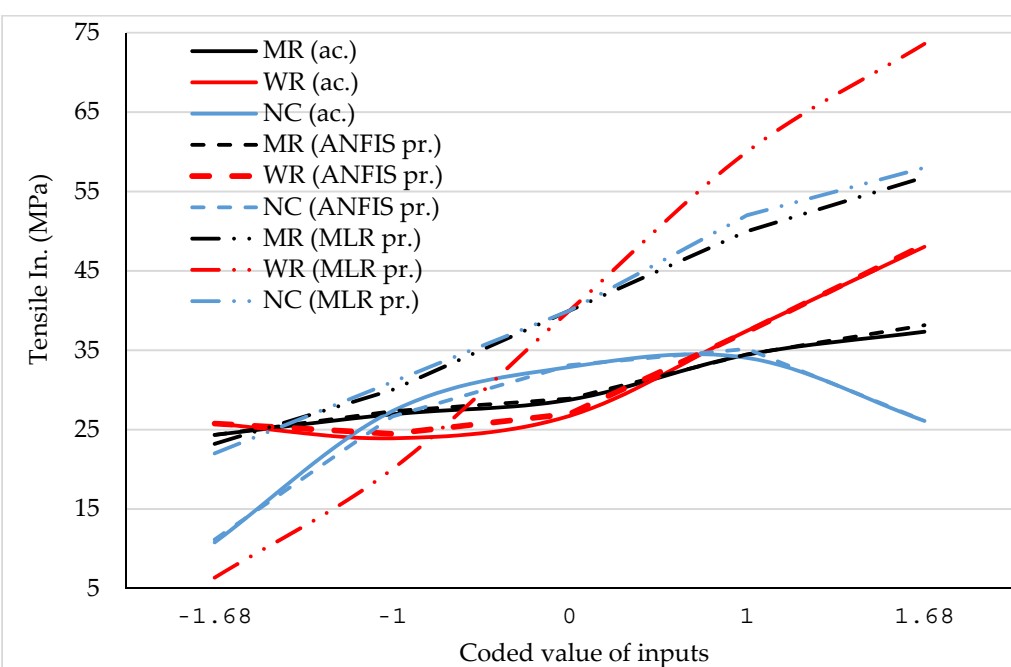

**Figure 9.** Comparison of the direct effect of independent variables on the actual tensile index values and those predicted by the ANFIS and MLR approaches.

As the starch increases, the UF-OS system viscosity decreases, because the hemiacetal ring of sugars opens and the glycoside bonds break resulting in the decrease in the polymer chain length. The decrease in viscosity leads to the increase in the flowability of the adhesive, increasing the resin penetration inside the paper [39]. However, as the temperature increases gradually and becomes the maximum, the adhesive viscosity increases to several times, where the adhesive containing more starch due to the more effective influence of starch to create cross-linking of the UF resin at a certain temperature when it is coagulated becomes more viscous [4]. It results in a stronger connection in the substrate as the starch to UF weight ratio increases and the adhesive penetration increases. Since the decrease in the F to U ratio decreases the coagulation rate and the cross-linked density [4], the increase in the modified starch to UF weight ratio plays the role of a cross-linked factor in the UF resin coagulation process.

As the MR increases, the methylene ($–CH_2–$) and methylol ($–CH_2OH$) bridges decrease. It is related to the high resin formaldehyde ratio and the increase in aldehyde and carboxyl groups of the modified starch in resin due to the less urea added. At first, urea may react with formaldehyde to form urea methylol due to the unstable ether connections while formaldehyde level has remained constant in resin. Urea reacts with the –CHO groups of the oxidized starch to form U-OS-based polymer. As the F to U molar ratio increases, methylene connection ($–CH_2$) increases and methylol groups ($CH_2OH$) decrease, indicating that methylene bonds ($–CH_2–$) are the dominant type, extremely conjugating oxidized starch and urea [40].

When treating starch with NaOH, some compounds with a low molecular weight are produced [41] which is due to the separation of the intermolecular hydrogen bonds in amylopectin [42]. During it, the rigidity and as a result, the stability of the molecular structure of starch decrease and amylose chains become more movable and the granule architecture declines [43]. The increase in the modified starch content decreases the flow time. It can be described by the fact that at higher starch levels, less time is needed for coagulation, because the presence of higher carbonyl functional groups accelerates the formation of chemical connections in the veneer. Due to the higher water absorption properties, the adhesive has a higher moisture and less coagulation time so that different pre-coagulation degrees of dried papers can be described by it. The higher final moisture

means that the paper is treated more mildly than papers with a lower final moisture when drying. Hence, the adhesive will bear less pre-coagulation time in the moister paper and the time to complete the cross-linking increases. However, according to the time spent to treat the papers thermally, it seems that the better penetration of adhesive in paper on one hand and the enough time of resin coagulation on the other hand have increased the tensile strength of paper. XRD studies have also shown that chemical treatment has changed and made special peaks small. However, the pattern of XRD curves of starch and modified starch is similar [1]. It means that the crystalline domain of the modified starch is damaged to some extent [2]. At the same time, in the combination of modified starch and UF, all peaks disappeared, because the modified starch and urea converted into OS-U adhesive [44]. In the TGA analysis, the modified resin has shown that it has a lower peak and initial temperatures, meaning that adding starch accelerates the UF resin coagulation. However, as the F to U molar ratio decreases, the resin coagulation decreases due to the decrease in the cross-linking density. The application of higher levels of starch plays the role of the cross-linked factor in the resin coagulation process. Hence, the decrease in the F to U molar ratio results in faster resin coagulation as the modified starch is added [40]. The weight loss of the starch and aminoplast adhesive is less than the aminoplast resin. Moreover, this combined system also has a higher thermal stability that is related to the formation of chemical connections between starch and aminoplast resins. It is due to the UF and starch chemical connection that has made the UF-OS resin stronger [4].

The increase in the storage modulus (G') indicating the elastic properties due to the structural connections of the 3D network of amylose and inflated granule composite system shows the increase in the formation of 3D networks and cross-linking of starch specimens [45]. By applying the treatment and increasing the hydroxyl groups gradually, the ability to contact the resin polymer and as a result, the grafting improve and the hydrogen bonds between starch chains decrease and the starch destruction process stops [40]. According to the loss modulus parameter (G", showing the viscous properties of the molecular network structure) affecting the loss tangent (tanδ), i.e., the ratio G"/G' [46], the treatment has decreased the starch aggregation and improved its distribution in the resin polymer to form better grafting reaction.

The starch morphology shows a granular structure with a smooth surface which is aggregated to some extent while the oxidized starch has a destroyed structure with rough surface profiles due to the starch oxidation reaction with oxidizer and the broken polymer chain [47]. The complete disappearance of the starch shape and its conversion into the bar shape in the OS-U adhesive indicates the occurrence of the polycondensation reaction between the oxidized starch and urea. Adding starch to aminoplast resins makes the surface shape show the interaction between two components so that as the starch increases, the specimens' surface becomes rougher and indicates ductile fracture, which is due to the covalence connections between starch and resin molecules improving the ductile fracture behavior [4]. Furthermore, the presence of the compact adhesive system can indicate that the interaction of nanoparticles and the OS-U adhesive has occurred [1,5].

$SiO_2$ is a more amorphous substance compared to the UF resin [11]. X-ray diffraction analyses showed that as the amorphism increased due to adding nano-silica to the UF resin, the peak width increased in the range $22°\theta$. Confirming the interaction between resin and nano-silica, the resin surface morphology analysis indicated that as silica nanoparticles were added to the UF resin matrix, more small bright regions emerged so that after performing the Energy-dispersive X-ray spectroscopy analysis, it became clear that the presence of these regions is the result of the presence of silica nanoparticles. On the other hand, as more nanoparticles were added to the matrix, the aggregation degree was strongly affected. Confirming the interactions between resin and silica nanoparticles, the analysis of the FTIR peaks results also showed that the peak position at 3350 cm$^{-1}$ has shifted to lower wave numbers, so that it can be concluded that due to the presence of silanol groups at the nano-$SiO_2$ surface that are very active, a hydrogen bond is formed or condensation reactions occur [11]. By improving the interaction between nano-silica and matrix, these

changes can be due to the complete coating of $SiO_2$ particles by resin so that silica porous spherical particles become smooth and form well. Based on the EDS, it was also indicated that no Si element has appeared in the product, indicating that $SiO_2$ microspheres are coated completely [48]. Adding starch results in an increase in the resin curing temperature due to the interaction of hydroxyl and carbonyl groups in starch with resin polymer chain so that a bond is created with it or in condensation reaction and as a result, the polymer chain needs more energy to form the cross-link and complete the curing process [33]. Since the presence of nanoparticles makes it possible to transfer temperature more quickly in the adhesive compound, the need to increase the temperature quickly due to the increase in the modified starch and the subsequent increase in the functional groups resulting from glucose rings will be met. Hence, the cross-link is supported more.

## 4. Conclusions

In the present research, the application of the MLR and ANFIS was examined to predict to the tensile index of the paper impregnated by the UF resin containing the modified starch adhesive. The following conclusions are the findings of the research:

1. The correlation analysis showed that there is a significant relationship between the tensile index of the paper impregnated by resin and the MR, WR and NC.
2. The comparison of the models produced to estimate the tensile index showed that $R^2$ was more in the ANFIS model while RMSE, MAE and SSE values were less in the ANFIS model. According to the statistics, the ANFIS model has showed a better performance to predict the response being examined compared to the MLR.
3. The starch modified by some functional groups including carbonyl, hydroxyl, etc., can react with the UF resin confirmed by FTIR spectroscope. Based on the TEM analysis, adding silica nanoparticles resulted in more matrix connection and its better distribution, but when increasing nanoparticles to the maximum level, the uniform distribution of nanoparticles was affected negatively.
4. Increasing nano-silica beyond the middle level, the tensile index increased continuously as the WR increased. Increasing the MR and NC to the middle level, the tensile index increased. The intensity of the effect of the increase in the modified starch consumption has been much more than that of the effect of MR and NC.

**Author Contributions:** Conceptualization, methodology, software, validation, formal analysis, investigation and resources, project administration, writing and original draft preparation, M.N. and H.R.K.; investigation, writing—review and editing writing and supervision, A.N.P., D.F. and E.V.; visualization, H.R. and H.K. All authors have read and agreed to the published version of the manuscript.

**Funding:** This research received no external funding.

**Conflicts of Interest:** The authors declare no conflict of interest.

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
