# Peer review of "Comparison of the Estimation Ability of the Tensile Index of Paper Impregnated by UF-Modified Starch Adhesive Using ANFIS and MLR"

_jcs, doi:10.3390/jcs6110341_

Round 1

Reviewer 1 Report

The paper presents an interesting approach based on the Comparison of the estimation ability of the tensile index of paper impregnated by UF-modified starch adhesive using ANFIS and MLR . However, the innovation of the current research work should be further highlighted and emphasized. At the same time, the authors should consider the following comments to greatly improve the quality of the paper.

1. In the abstract, add a final statement that highlights the importance of this research and its possible potentials.

2. The introduction needs to be improved by relating to the mechanics of the studied materials and their mechanical characteristics. The references to be included are: 10.1177/0021998318790093, 10.1016/j.polymertesting.2017.09.009, 10.1016/j.compstruct.2021.114698, 10.1177/0731684417727143, 10.1002/app.46770, 10.1016/j.porgcoat.2022.107015.

3. Kindly add a table that describes the main physical and chemical properties of the raw materials used in this study.

4. Were the preparation methods described by the authors come in accordance with a certain standard or do they follow previous procedures?

5. For the tensile tests, what was the reason for the specified test conditions in this research? Do the speed of test value relate to a specific application? 

6. How many samples were used per configuration for the tensile test?

7. Why there is a low correlation factor in both diagrams of figure 4 between actual and predicted values obtained by the two models? Have you tried using other models for training?

8. The conclusion needs to be modified to summarize the research outcomes in short statements with clear observations.

Author Response

At first I have to say that reviewer is very exact and this manuscript has been improved with helping reviewers. It is very fortune for me. Hence, we appreciate you for careful and thorough reading of this manuscript. We revised this manuscript according to your helpful and constructive suggestions and added new explanations to the revised version of the manuscript. Here are our answers to your specified questions

Comments and Suggestions for Authors

The paper presents an interesting approach based on the Comparison of the estimation ability of the tensile index of paper impregnated by UF-modified starch adhesive using ANFIS and MLR . However, the innovation of the current research work should be further highlighted and emphasized. At the same time, the authors should consider the following comments to greatly improve the quality of the paper.

  1. In the abstract, add a final statement that highlights the importance of this research and its possible potentials. Now, I think that is better.
  2. The introduction needs to be improved by relating to the mechanics of the studied materials and their mechanical characteristics. The references to be included are: 10.1177/0021998318790093, 10.1016/j.polymertesting.2017.09.009, 10.1016/j.compstruct.2021.114698, 10.1177/0731684417727143, 10.1002/app.46770, 10.1016/j.porgcoat.2022.107015.

I added fallowing references as I could:

  1. Zaghloul, M.M.Y.; Mohamed, Y.S.; El-Gamal H. Fatigue and tensile behaviors of fiber-reinforced thermosetting composites embedded with nanoparticles. J. Compos. Mater. 53 (6), 2018, 709-718. https://doi.org/10.1177/0021998318790093

22. Zaghloul, M.M.Y.; Zaghloul, M.Y.M.; Zaghloul M.M.Y. Experimental and modeling analysis of mechanical-electrical behaviors of polypropylene composites filled with graphite and MWCNT fillers. Polym. Test. 63, 2017, 467- 474. https://doi.org/10.1016/j.polymertesting.2017.09.009

15. Zaghloul, M.Y.M.; Zaghloul, M.M.Y.; Zaghloul, M.M.Y. Developments in polyester composite materials – An in-depth review on natural fibres and nano fillers. Compos. Struct. 278, 2021, 114698. https://doi.org/10.1016/j.compstruct.2021.114698

  1. Kindly add a table that describes the main physical and chemical properties of the raw materials used in this study.

I tried to describe

  1. Were the preparation methods described by the authors come in accordance with a certain standard or do they follow previous procedures?

According to my some previous studies about preparation of Modified starch and UF resin.

  1. For the tensile tests, what was the reason for the specified test conditions in this research? Do the speed of test value relate to a specific application? 

 Yes. Generally, in wood and wood products area, the samples have to be conditioned in special situation for example at relative humidity=65% and temperature of 21°C to certain for stabilization of samples. Besides, according to standards, the time for create fracture has to be 120s±30s. according to this time, the speed of loading is regulated.   

  1. How many samples were used per configuration for the tensile test?

As I mentioned in the text, the number of replicate for each sample was 3. The total of treatment was 15 with respect to the used experimental design (central composite rotatable design, CCRD).

  1. Why there is a low correlation factor in both diagrams of figure 4 between actual and predicted values obtained by the two models? Have you tried using other models for training? As I described in text, there is excellent corolation between predicted and experimental values according to ANFIS approach. Of course the is poor correlation between predicted and actual values according to MLR approached. R2 shows this differences. At the end, probably some information was omitted or … Now, I modified modified text in this figure.
  2. The conclusion needs to be modified to summarize the research outcomes in short statements with clear observations. I tried to do it.

We thank the reviewer for this valuable suggestion. We revised this manuscript according to your suggestion.

Once again, thank you for your valuable comments.

Reviewer 2 Report

This manuscript by Nazerian et.al., titled “Comparison of the estimation ability of the tensile index of paper impregnated by UF-modified starch adhesive using ANFIS and MLR” presents a very interesting study about predicting the strength of paper-based veneer using MLP and ANFIS. The manuscript is very well written and presented. It needs a few minor changes as listed below:

 ·        Page 1, Line 16, “…to UF resin (WR) containing different…” : The authors have used the acronym “UF” here without prior explanation of what that means. Please mention the full form on the first instance for every word that is mentioned as an acronym in the text. Same with “FTIR” on Page 1, Line 20 and “ANNs and ANFS” on Page 2, Line 71.

·        Page 1, Line 20, “FTIR Spectroscopy and Transmittance Electron Microscopy…”: It is well known as Transmission Electron Microscopy and the authors needs to mention that.

·        Page 1. Line 34,  “Two saturation and impregnated stage…: The first paragraph seems to be out of place in the introduction. It does not introduce the research topic in hand. It rather starts with literature survey. Please use this paragraph later in the introduction and re edit the first few paragraphs of the introduction section to have a good flow of text for the readers.

·        Page 10, Figure 2: Please show bigger font size of the scale shown. A bigger scale in white might look brighter.

The rest of the manuscript including the Materials and methods, results and discussion, conclusion, references, and others, look well done with proper sentence constructions. The scientific logic of the manuscript flows very well in conjunction with the current literature

Author Response

At first I have to say that reviewer is very exact and this manuscript has been improved with helping reviewers. It is very fortune for me. Hence, we appreciate you for careful and thorough reading of this manuscript. We revised this manuscript according to your helpful and constructive suggestions and added new explanations to the revised version of the manuscript. Here are our answers to your specified questions

This manuscript by Nazerian et.al., titled “Comparison of the estimation ability of the tensile index of paper impregnated by UF-modified starch adhesive using ANFIS and MLR” presents a very interesting study about predicting the strength of paper-based veneer using MLP and ANFIS. The manuscript is very well written and presented. It needs a few minor changes as listed below:

  • Page 1, Line 16, “…to UF resin (WR) containing different…” : The authors have used the acronym “UF” here without prior explanation of what that means. Please mention the full form on the first instance for every word that is mentioned as an acronym in the text. Same with “FTIR” on Page 1, Line 20 and “ANNs and ANFS” on Page 2, Line 71.

They were corrected

  • Page 1, Line 20, “FTIR Spectroscopy and Transmittance Electron Microscopy…”: It is well known as Transmission Electron Microscopy and the authors needs to mention that. It was corrected
  • Page 1. Line 34,  “Two saturation and impregnated stage…: The first paragraph seems to be out of place in the introduction. It does not introduce the research topic in hand. It rather starts with literature survey. Please use this paragraph later in the introduction and re edit the first few paragraphs of the introduction section to have a good flow of text for the readers.

It was corrected

  • Page 10, Figure 2: Please show bigger font size of the scale shown. A bigger scale in white might look brighter.

It was done

 The rest of the manuscript including the Materials and methods, results and discussion, conclusion, references, and others, look well done with proper sentence constructions. The scientific logic of the manuscript flows very well in conjunction with the current literature

We thank the reviewer for this valuable suggestion. We revised this manuscript according to your suggestion.

Once again, thank you for your valuable comments.

Round 2

Reviewer 1 Report

The authors have partially fulfilled the comments. Kindly enrich the introduction by relating the mechanics to these references: 10.1177/0731684417727143, 10.1002/app.46770, 10.1016/j.porgcoat.2022.107015.

Author Response

At first I have to say that reviewer is very exact and this manuscript has been improved with helping reviewers. It is very fortune for me. Hence, we appreciate you for careful and thorough reading of this manuscript. We revised this manuscript according to your helpful and constructive suggestions and added new explanations to the revised version of the manuscript. Here are our answers to your specified questions

Referee 1:

The authors have partially fulfilled the comments. Kindly enrich the introduction by relating the mechanics to these references: 10.1177/0731684417727143, 10.1002/app.46770, 10.1016/j.porgcoat.2022.107015.

These references were added.

Thanks a lot for your comments.

Reviewer 2 Report

This manuscript by Nazerian et.al., titled “Comparison of the estimation ability of the tensile index of paper impregnated by UF-modified starch adhesive using ANFIS and MLR” looks much better now after the first revision. It needs one minor change, which I wish the authors would have already worked on, in the first round of revision. It is listed below:

The font on the scale in Fig 2b is still unclear. The authors need to use some other font color like white for both the images. Increasing the font size will also make it better.

Author Response

At first I have to say that reviewer is very exact and this manuscript has been improved with helping reviewers. It is very fortune for me. Hence, we appreciate you for careful and thorough reading of this manuscript. We revised this manuscript according to your helpful and constructive suggestions and added new explanations to the revised version of the manuscript. Here are our answers to your specified questions

Referee 2

The font on the scale in Fig 2b is still unclear. The authors need to use some other font color like white for both the images. Increasing the font size will also make it better.

Thanks a lot. Now, I thinks that it is OK.

Round 3

Reviewer 1 Report

The authors have succeeded to fulfill the requirements to improve their manuscript. The paper can be accepted for publication.

Reviewer 2 Report

The manuscript now looks ready to be published